# Augmenting Evolutionary Models with Structure-based Retrieval

Yining Huang [* 1]  Zuobai Zhang [* 2 3]  Jian Tang [2 4 5]  Debora S. Marks [1 6]  Pascal Notin [1]

## Abstract

Multiple Sequence Alignments (MSAs) are crucial in protein sequence analysis for identifying homologous proteins sharing a common evolutionary origin. However, traditional MSA search tools struggle to recover distantly related sequences that, despite low sequence similarity, exhibit high structural and functional resemblance—often missing in the so-called 'midnight zone' of protein similarity. To overcome these limitations, we propose the integration of structure similarity search tools to enhance the identification of homologous proteins. This approach utilizes Foldseek to search the AlphaFold database, aligning structurally similar proteins to construct Multiple Structure Alignments (MStructAs) alongside traditional MSAs. By combining these alignments, we develop family-specific generative models for protein fitness prediction, using diverse assays from the ProteinGym benchmarks. Our findings reveal that incorporating structure-based retrieval into MSAs significantly improves the performance of alignment-based methods, suggesting a robust hybrid retrieval strategy that harnesses both sequence and structure similarities.

## 1. Introduction

Multiple Sequence Alignments (MSAs) have long served as fundamental tools for protein sequence analysis, aiming to retrieve homologous proteins from large databases. These homologous proteins belong to the same family and share a common evolutionary origin, reflected by their sequence, structure and function similarity. By leveraging the signal of evolution among the homologous sequences,

these tools have been used to analyze the conservation of single sites in protein (Adzhubei et al., 2010; Hecht et al., 2015; Huang et al., 2017; Kircher et al., 2014). Building on protein sequence alignments, researchers have also developed methods for predicting protein structures (Jumper et al., 2021; Abramson et al., 2024), inferring the expression and activity of protein domains (Frazer et al., 2021; Laine et al., 2019), and learning protein representations (Truong Jr & Bepler, 2024; Rao et al., 2021).

Existing MSA search tools recover homologous sequences primarily based on sequence similarity considerations (Altschul et al., 1990; 1997; Steinegger & Söding, 2017). However, these approaches typically fail in recovering distantly related sequences with high structure and functional similarity but low sequence similarity, referred to as the 'midnight zone' of protein similarity (Heinzinger et al., 2021). To address the limitations of traditional Multiple Sequence Alignments (MSAs), we explore the usage of structure similarity search tools to efficiently identify homologous proteins with similar structures from large databases. The proteins identified are then aligned with the target protein to create Multiple Structure Alignments (MStructAs), which can uncover insights into the "blind spots"of standard sequence-based MSAs. To demonstrate the effectiveness of our methods, we use 197 Deep Mutational Scanning (DMS) assays from ProteinGym (Notin et al., 2023) as examples. For each target protein, we employ Foldseek to search for structurally similar proteins in the AlphaFold UniProt database and align them to construct MStructAs. We then combine the identified MSAs and MStructAs to train family-specific generative models for protein fitness prediction (Frazer et al., 2021). Our results indicate that the performance of these alignment-based approaches can be markedly improved by augmenting MSAs with sequences recovered via structure-based retrieval, paving the way for hybrid retrieval strategies that consider both sequence and structure similarities.

## 2. Method

Inspired by recent development of fast protein structure tools, in this section, we explore the usage of Multiple Structure Alignments (MStructAs) as a complement of Multiple Sequence Alignments (MSAs). Specifically, we illustrate

*Equal contribution [1]Harvard Medical School [2]Mila - Québec AI Institute [3]Université de Montréal [4]HEC Montréal [5]CIFAR AI Chair [6]Broad Institute. Correspondence to: Yining Huang <yininghuang@hms.harvard.edu>, Zuobai Zhang <zuobai.zhang@mila.quebec>, Pascal Notin <pascal_notin@hms.harvard.edu>.

*Accepted at the 1st Machine Learning for Life and Material Sciences Workshop at ICML 2024.* Copyright 2024 by the author(s).

the idea on the 197 wild type proteins from the ProteinGym benchmarks as example. We introduce the process of generating MStrcutA for a target protein (Sec. 2.1) and show the complementary effect of MStructA on MSA (Sec. 2.2).

## 2.1. Multiple Structure Alignment

**MStructA Searching** To construct MStructA for each wild-type protein in ProteinGym, we utilize Foldseek (van Kempen et al., 2024) to search for structurally similar proteins within the AlphaFold Database (Varadi et al., 2024). Foldseek not only retrieves protein structures but also provides the corresponding sequence alignments with the target protein. This allows us to seamlessly concatenate the sequence alignments from MStructA with the MSA for downstream applications. To ensure sufficient number of retrieved structures, we set the amount of prefilter handed to alignment in Foldseek to 50,000 and keep other configurations as default. This approach enables us to identify the broadest possible spectrum of structural homologs.

It is important to note that the original multiple sequence alignments (MSAs) for each protein in ProteinGym were searched against UniRef100 (Suzek et al., 2007), a non-redundant subset of the UniProt database (Consortium, 2022). With the aim to discover previously unrecognized homologs, we select the AlphaFold UniProt database for structure alignment searches, which offers predictions for over 200 million protein structures of UniProt sequences. This potentially provides a valuable complement to traditional MSAs and enhancing our understanding of protein structure and function.

**MStructA Filtering** As we enhance the recall rate by increasing the number of prefilters in Foldseek, the retrieved MStructA above may include protein sequences of low quality and relevance. To balance the trade-off between the recall and precision of retrieved proteins, we follow the practice of previous alignment-based models (Frazer et al., 2021; Hopf et al., 2017; Riesselman et al., 2018). Specifically, we only keep sequences with Sequence Identity $>0.1$, E-value $<1e-10$, and Gaps in Sequence $<50\%$. This ensures that the MStructA includes only high-quality structural homologs.

## 2.2. Complementary Effect of MStructA on MSA

To study the overlap between MSA and MStructAs, we combine the multiple sequence alignment (MSA) and multiple structure alignment (MStructA) together, and remove duplicated sequences in MSA and MStructA. We find that most alignments shows less than $5\%$ depth increase. Nevertheless, it is interesting that there are still some alignments have more than $200\%$ depth increase. It might implies the original MSA failed to recover many potentially functional and evolutionary relevant proteins.

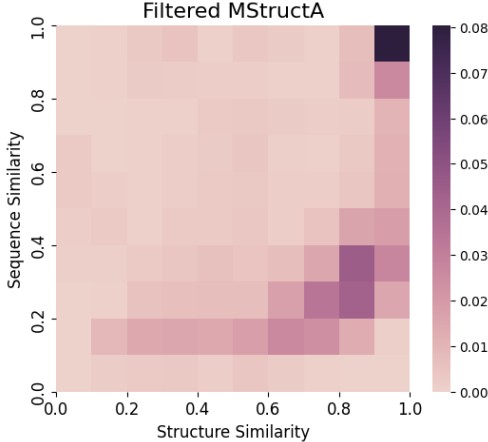

*Figure 1.* Heatmap for the distribution of sequence and structure similarity scores for MStructA proteins across different assays, with each cell colored according to the percentage of proteins falling into that sequence and structure similarity score bin.

To determine if MStructA uncovers previously unknown proteins within MSA, we calculate both sequence similarity (sequence identity) and structural similarity (TM-score) for each protein in MStructA compared to their corresponding target protein. We categorize the sequence and structure similarity scores into 10 discrete bins and compute the percentage of proteins in each bin for every assay. The average results across all assays are plotted as a heatmap in Figure 1. Our analysis reveals that most proteins identified by MStructA exhibit high structural similarity but low sequence similarity to their targets. This suggests that MStructA effectively supplements the MSA by identifying structural homologs that were not previously detected.

## 3. Experiments

To further evaluate the complementary effect of MStructA on MSA, in this section, we focus on the application of MStructA on protein fitness prediction tasks.

### 3.1. Experiment Setup

**Datasets.** We use ProteinGym, a comprehensive collection of datasets and models for protein fitness prediction and design, to benchmark our method. It contains protein fitness data from deep mutational scanning (DMS) experiments, providing detailed effects of mutations on protein function.

We aim to study how multiple structure alignments (MStructA) improve the performance of fitness prediction models when they complement multiple sequence alignments (MSA). Thus, we selected 30 assays with more than $5\%$ overall depth increase when MStructA is combined with MSA. The selected assays have varying original MSA

*Table 1.* Overall results for EVE trained with only MSA and EVE trained with combined MSA and MStructA.

|  | SPEARMAN | AUC |
| --- | --- | --- |
| EVE (MSA) | 0.434 | 0.737 |
| EVE (MSA+MSTRUCTA) | **0.443** | **0.742** |
| % ASSAYS IMPROVED | 60% | 61% |

depths, protein families, and functional types.

**Model.** We use a zero-shot alignment-based model for our experiment because supervised models require functional annotations, which can often be sparse, biased, and low-quality. Zero-shot models have the advantage of predicting protein fitness without requiring extensive labeled training data. These alignment-based models leverage the evolutionary relationships learned from MSAs to infer the functional impact of mutations to predict protein fitness.

Specifically, we choose EVE (Frazer et al., 2021), an alignment-based model that has shown good performance using multiple sequence alignments (MSA) for protein fitness prediction. It models the natural distribution of protein sequences resulting from evolutionary processes to capture the constraints that maintain protein fitness. This allows it to estimate fitness changes for any variant by comparing the relative likelihoods of different sequences.

**Training.** For each assay, we train an EVE model on the combined MSA and MStructA. We also set all columns in the alignment focus positions when training EVE with combined MSA and MStructA. This makes training consistent with the EVE trained on MSA in ProteinGym, allowing a fair comparison. Each model is trained for 400,000 steps with a learning rate of 0.0001 and a batch size of 256. Training time varies between assays due to different protein sequence lengths. The average training time for an EVE model is around 16 hours on an A100 GPU.

### 3.2. Performance

Following ProteinGym, we evaluate the performance of our method by calculating Spearman correlation and Area Under the ROC Curve (AUC) for each assay. We compare our method of training EVE with combined MSA and MStructA against the original EVE with only MSA. The average results across all assays are reported in Table 1.

The results show that our method outperforms the original EVE using only MSA in both metrics. We also calculate the percentage of assays showing improvement compared to the original EVE using only MSA. Around 60% of assays show improvement in Spearman and AUC when using our method. This finding demonstrates that incorporating high-quality structural similar sequences can provide meaningful evolutionary information important for fitness prediction. By leveraging the additional structural evolutionary context provided by MStructA, our method achieves a more comprehensive understanding of protein fitness, leading to better overall performance.

### 3.3. Breakdown Analysis

We group assays into low depth, medium depth, and high depth based on their original MSA depth. We then calculate the average Spearman of our method and original EVE for each group (Table 2). The result shows that our method has the largest improvement in assays with a low original MSA depth. The low-depth MSAs may not contain enough structurally informative sequences for the model to effectively produce a natural distribution of protein sequences. The MStructAs complement these MSAs by providing additional structurally similar protein sequences, so that the model can better capture the evolutionary relationship between protein sequences.

We also find that our methods exhibit the largest performance improvement on assays of the binding function type. EVE trained on combined MSA and MStructA outperform original EVE with only MSA by an average of 0.066 for assays of binding function type (Table 3). Binding functions inherently rely on structural information, as the precise 3D arrangement of amino acids is crucial for the interaction between proteins and their binding partners. This finding implies that MSA built on sequence similarity only might not fully capture the structural context essential to accurately infer these interactions. The incorporation of MStructA helps the model learn the relationship between proteins in both the sequence and structure context. This integration of structural information is particularly beneficial for predicting the fitness of proteins involved in complex structural interactions, ensuring a more comprehensive understanding of the fitness landscape.

## 4. Discussion and Future work

**MStructA Quality** For several protein families, the current filtering criteria remove too many potentially-relevant sequences, resulting in many MStructAs having a very low depth. Future work will focus on optimizing our filtering pipeline to balance the quality of included sequences with the number of retrieved sequences to provide optimal structural information gain. Eventually, this pipeline should be broadly applicable, and strike that optimal balance out-of-the-box across all protein families and experimental assays.

Since the presence of too many gaps in the alignments will degrade the performance of evolutionary models, our filters remove 'fragment' sequences with more than 50%

*Table 2.* Results grouped by original MSA depths for EVE trained with only MSA and EVE trained with combined MSA and MStructA.

| ORIGINAL MSA DEPTH | EVE (MSA) | EVE (MSA+MSTRUCTA) | IMPROVEMENT |
|---|---|---|---|
| LOW | 0.497 | 0.534 | **0.037** |
| MEDIUM | 0.425 | 0.424 | -0.001 |
| HIGH | 0.349 | 0.363 | 0.014 |

*Table 3.* Results grouped by function types for EVE trained with only MSA and EVE trained with combined MSA and MStructA.

| FUNCTION TYPE | EVE (MSA) | EVE (MSA+MSTRUCTA) | IMPROVEMENT |
|---|---|---|---|
| ACTIVITY | 0.436 | 0.441 | 0.005 |
| BINDING | 0.444 | 0.510 | **0.066** |
| EXPRESSION | 0.450 | 0.462 | 0.012 |
| ORGANISMALFITNESS | 0.426 | 0.431 | 0.005 |

gaps to ensure the robustness and reliability of the model. However, many structurally similar sequences may have high sequence divergence, resulting in more than 50% gaps. To address this, we need to develop better MStructA construction methods that can include such structurally similar but sequence-dissimilar proteins. Alternatively, we could explore models that are agnostic to gaps or can utilize MStructA in formats other than sequence alignments. This will allow us to leverage the structural information more effectively and further improve the performance of evolutionary models.

**Benchmarking** Due to limited computational resources, we benchmarked only 30 randomly selected assays that shows more than 5% increase in overall depth from MSA to combined MSA and MStructA. Additionally, our current filtering criteria result in many assays exhibited less than 5% depth increase. Benchmarking these assays would not yield meaningful insights, as the structural context gain is minimal. Once we develop a MStructA construction pipeline capable of building high-quality MStructAs for all assays while balancing depth and quality, we will conduct comprehensive benchmarking across all assays in ProteinGym.

Furthermore, we plan to benchmark other alignment-based models (eg., PSSM, Potts (Hopf et al., 2017)), or hybrid models such as TranceptEVE (Notin et al., 2022), which combines the strengths of family-specific alignment-based methods and family-agnostic language models. This will provide a broader evaluation of our method and its applicability to various frameworks.

Lastly, to facilitate the broader use of these hybrid sequence-based and structure-based MSAs by practitioners, we aim to develop unified protein retrieval packages to do the combined search via a common interface and process.

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
