# OpenReview forum: "Augmenting Evolutionary Models with Structure-based Retrieval"
_ICML.cc/2024/Workshop/ML4LMS — ML4LMS Poster_

### Official Review · Reviewer_vD16 · 2024-06-10
**It is a good starting point for future research**

**Rating:** 6
**Confidence:** 4

**Review:**

- the paper is well written and concise
- it addresses and interesting take in augmenting MSA with MStructAs
- it shows improvements when training EVE on MSA alone or the MStructAs

Feedback for future work:
- potentially would be good to check whether the model is imroving just because you are providing it more sequences, for this you could try testing adding sequences of varying MStructAs as training and see the impact on the model performance
- given the relatively small differences in performance, boostrapping and potentially confidence intervals could be measured to ensure the validity of the results not to be attributed to noise.

---

### Official Review · Reviewer_J8yL · 2024-06-11
**Structure-based Retrieval for protein fitness prediction, results are preliminary but seem promising**

**Rating:** 6
**Confidence:** 3

**Review:**

This work proposes to incorporate Multiple Structure Alignments (MStructAs) and Multiple Sequence Alignments (MSAs) for protein fitness prediction. The authors use structure similarity search tools to find structurally similar proteins and build MSructsAs that are then combined with MSAs and used to train alignment-based generative models for fitness prediction. The integration of structural information appears to hold promise and help improve performance in some of the performed experiments.


**Pros**
- The paper is overall well-written and easy to follow. The results are preliminary but seem promising.
- The proposed approach is well-motivated.

**Cons**
- The paper is missing some details about the training and inference of the generative models.